# Facts and Myths about GM Food—The Case of Poland

**Paulina Kubisz** [1], **Graham Dalton** [2], **Edward Majewski** [3,*] **and Kinga Pogodzińska** [3]

1  Faculty of Economics, MBA in Agribusiness Management, Warsaw University of Life Sciences, Nowoursynowska 166, 02-787 Warszawa, Poland; paulinakubisz83@gmail.com
2  Formerly of Aberdeen University and the Scottish Agricultural College, Aberdeen AB24 3FX, UK; graham@contlaw.com
3  Institute of Economics and Finance, Warsaw University of Life Sciences, Nowoursynowska 166, 02-787 Warszawa, Poland; kinga.pogodzinska@wp.pl
*  Correspondence: Edward_majewski@sggw.edu.pl; Tel.: +48-225-934-216

**Abstract:** The importance of biotechnology for the global economy is growing, including developments in the field of genetically modified organisms (GMO), which have revolutionized the cultivation of several major food crops. Despite the many benefits from introducing genetic modifications to crops, the Polish society shows a strong distrust towards GMO-based food. The negative attitude of the society towards genetically modified (GM) food could be considered irrational. It is not supported by adequate knowledge and is based on fears, despite the fact that there is no scientific evidence of threats of GM products towards the environment, health, or human life. Details of these perceptions were revealed within Polish society from surveys of three groups of respondents: consumers, students, farmers. Data from the surveys have been compared with the answers to the same questions by five biotechnology experts from Polish academic institutions. A general observation from the analysis of the survey results and past studies quoted in the literature review is that the level of understanding and acceptance of GMO technologies is still low in Polish society, and, to a large extent, is based on stereotypes rather than on scientific knowledge. They show broad support for the general benefits of GMOs, which does not vary between the three groups of respondents surveyed, but noticeably differs with the experts' views. GMOs have allies, but also opponents who have their own beliefs shaped largely by unreliable information disseminated through the Internet and social media. Providing more reliable targeted information on GMOs based on scientific evidence can have an important role in changing polarized attitudes towards GM food.

**Keywords:** GMO; genetically modified food; public attitude towards GM food



## 1. Introduction

Genetically modified organisms (GMO), currently almost entirely crop plants, since their commercial introduction in 1996 [1] remain a recurrent issue within the forums of contemporary debate. The stakeholders consisting of the scientific community, farmers, the processing industry, and consumers, augment technological concerns with anticipated social, economic, and political consequences. Such attention is largely focused on genetically modified (GM) foods, which are defined by the World Health Organization (WHO) as "foods derived from organisms whose genetic material Deoxyribonucleic acid (DNA) has been modified in a way that does not occur naturally, e.g., through the introduction of a gene from a different organism" [2].

Consumers are extremely concerned about GM food safety, despite the existing regulations imposed on GM foods before allowing them onto the market, which are in accordance with the Food and Agriculture Organization (FAO)/WHO Codex framework for the safety assessment of GM food [3], as well as the lack of scientific evidence of risks to human health. Permission to introduce a GMO product onto the market requires analysis of the allergenic, teratogenic, and toxic potential of the modifications and antibiotic resistance. According

to numerous studies, the widely held opinions about the harmfulness of GMO varieties are groundless, and GMO food can be considered to be the most thoroughly tested in the world [4–11]. The legal aspects of the manufacturing, marketing, and control of genetically modified food and feed are also presented in several other publications, e.g., [12–15].

Beliefs regarding the harmfulness that GM foods represent arise largely from ideology, inertia, and ignorance about GMO procedures and perceived consequences, as contained in many negative campaigns by various anti-GMO groups [16–21]. An interesting observation may be made that the information against GM food appears in "waves". Whenever the GMO issue is brought to the political agenda, as periodically happens in the EU, immediately, different independent organizations—environmental, consumers associations, and individuals opposing the use of GMOs for food production become highly active [22]. Each source bombards public opinion with "arguments" that are often based on imprecise or even false information, but presented in a convincing way to fit specific prior beliefs.

An enhanced understanding of what GMOs really are and how they are used in practice is a key pre-condition for creating a rational attitude among consumers for accepting GM food. Since, as yet, no evidence has been produced regarding the negative, scientifically proven impacts of biotechnology and genetic modifications of crops on human health, the question "Where do all the society's concerns come from?" arises. This question is the basis of the paper. We examine hypotheses about the attitudes of Polish consumers towards GM foods and the associated lack of reliable sources of knowledge about genetic engineering procedures.

### 1.1. Genetically Modified Food: For and Against

Genetically modified food is the food that contains or consists of GMOs (such as bread made of GM wheat, soya beans, or canned corn) or is produced from GMOs (in whole or in part) but does not contain any DNA that has been changed. Genetic modifications are made through genetic engineering, which is one of the techniques used in biotechnology that involves a specific change in the genetic material (within a specific part of DNA) in order to eliminate an undesirable characteristic or to give a particular desired feature. This term emerged in the 1970s, when it became possible to transmit targeted genes (a gene is a part of DNA that contains information about the construction and function of a particular protein) through the pioneering work attributed mainly to three American biologists—Paul Berg, Stan Cohen, and Herbert Boyer [23]. They proved that the fundamentals of life are recorded in all organisms in the same way—through a genetic code. The introduction of a specific gene from one organism into another (taken from a different species) leads to the creation of a "transgenic organism". The transferred gene is called a "transgene", hence the name "transgenic organisms" or genetically modified organisms (GMOs). After the transfer, the transgene is permanently incorporated into the genome of the host across the boundaries between the different species.

The indisputable achievements of biotechnology in the pharmaceutical sector opened a path for other applications of gene transfer technologies, including the cultivation of crops and food production. Products of modern biotechnology have great potential. Plant resistance to difficult growth conditions (such as drought, soil salinity, cold, and insects) and potentially higher yields, compared with conventional methods of cultivation, could solve, at least in part, the problem of hunger in under-developed and developing countries. Moreover, higher yields could potentially allow for the reduction, at least at a local scale, of the area of cultivation, the use of water, and the inputs of fertilizers and plant protection chemicals. Many authors indicate that the resistance of GM crops to insects and herbicides reduces crop losses, leading to increased yields produced at lower costs [24–28] and a reduced use of polluting insecticides [11]. In general, genetic modifications of crops that are the basis for food production can make a significant contribution to the realization of several Sustainable Development Goals, mainly "zero hunger" and "good health and well-being".

Acker et al. [29], in their extensive review of the literature on the topic, indicate other positive effects of adopting GM crops, referring to numerous studies, such as:

- Growing adoption of no tillage practices in farming, which result in reduced inputs of fossil fuels, thus lowering farming's carbon footprint.
- The mitigation of soil erosion, which increases soils' productivity potential.
- The reduction of pesticide applications, which reduces farmers' exposure to chemicals, also lowering pesticide residues in food and feed crops, and releasing less chemicals into the environment.
- The possibility to reduce the level of mycotoxins in food and feed crops due to improved pest management.

Genetic modifications of plants may also result in better flavor, appearance, or chemical composition of crop products. Genetic modifications also slow fruit and vegetable maturation, which reduces losses in storage and transportation [30]. Other modifications enrich the product with vitamins, such as A, C, and E, and nutrients, e.g., unsaturated fatty acids, food cellulose, and probiotics [31].

The most recently introduced GM crops include non-bruising, non-browning, and late blight resistant potatoes; the first insect resistant (IR) sugarcane planted in Brazil; the first drought tolerant sugarcane (Indonesia); high oleic acid canola; isoxaflutole herbicide tolerant (HT) cotton; and HT and salt tolerant soybeans. All provide "various economically-important and nutritional quality traits beneficial to food producers and consumers in developing countries" [32].

Despite the many benefits from adopting GMO technologies for crop production, it is much more difficult to formulate well documented arguments against them. Outcrossing may be considered one of the most important threats, due to the risk of mixing genes from specific GMO plants with conventional crops and adjacent flora [29]. Paull [33], who characterizes GMO crops as "invasive species", provides examples of contaminations from GM canola in Australia, pointing out that they may cause particularly severe consequences for organic farms.

The productivity and economic aspects of the cultivation of GMO crops is another dimension in the discussion about GMO food, which are important for both consumers and food producers. Undoubtedly, genetic modifications of plants result in noticeable increases in production. According to Brookes and Barfoot [34], adopting GMO technologies in crop production in the years 1996–2013 resulted in additional production of 138 mln tons of soybeans, 274 mln tons of corn, 21.7 mln tons of cotton, and 8 mln tons of canola.

Several studies report better economic returns at the farm level, resulting mainly from higher yields and lower production cost of GM crops due to the reduced input of pesticides [1,35–37], but also from better crop quality and reduced labor inputs [38]. In some countries (e.g., in South America), an additional gain has been from sowing a second crop in the same growing season [1], such as soybeans, due to the shorter growth cycle of GMO wheat.

In general, it may be concluded that adopting GMO in crop production is not only profitable for individual farmers, but on average, is beneficial for the whole agricultural sector of those countries where GM crops are cultivated [1,39], as well as for the economies of developing countries [32,39].

In the 23rd year of the commercialization of biotech/GM crops in 2018, 26 countries grew 191.7 million hectares of biotech crops—an increase of 1.9 million hectares (4.7 million acres) or 1% from 189.8 million hectares in 2017. Except for the year 2015, this is the 22nd consecutive yearly increase. In 12 of the 18 years, double-digit growth rates of the areas with GMO crops was achieved.

The average biotech crop adoption rate in the top five biotech crop-growing countries increased in 2018 to almost all the cultivated crop areas in the USA (93.3%; average for soybeans, maize, and canola adoption), Brazil (93%), Argentina (~100%), Canada (92.5%), and India (95%). The expansion of biotech crop areas in these countries was fostered by immediate approval mechanisms and commercialization of new biotech crops and traits to target problems related to climate change and the emergence of new pests and diseases.

Soybeans lead at 95.9 million hectares, at 50% of the global biotech crop adoption, a 2% increase from 2017. This is followed by maize (58.9 million hectares), cotton (24.9 million hectares), and canola (10.1 million hectares). Based on the 2017 FAO global crop area for individual crops, 78% of soybeans, 76% of cotton, 30% of maize, and 29% of canola were biotech crops in 2018 [32].

### 1.2. Social and Policy Perspectives

While the range of benefits from the adoption of GMOs in agricultural production are quite obvious and well documented, it is difficult to debate fears related to GM crops and GM food, because suppositions regarding unknown long-term effects and safety concerns are hard to contest. The uncertainty comes from the fact that it is impossible to predict all of the implications of the long-term consumption of GM food.

More, and much stronger, drawbacks are perceived by consumers; however, these are based on beliefs created by information that lacks any solid scientific foundation. As an example, Jurkiewicz and Bujak [40], referring to studies conducted on rats, state that the consumption of GMO foods increases allergies and cancers. However, at the same time, they admit these results cannot be linked to GMOs. Similarly, Barrell [11] confronts fears of the potential of GM foods to trigger allergic reactions, beliefs that GM food may contribute to the development of cancer, or a threat that the resistance of some GMO plants to certain antibiotics could be passed on to humans, with statements from such organizations as The World Health Organization (WHO) or The American Cancer Society (ACS) arguing that there is no such scientific evidence.

Fears that foreign genes present in GM organisms can be transferred to pathogenic bacteria or even be incorporated into the human genome are sometimes overwhelming. The health issues seem to be the main concern influencing societal attitudes and political debate. In the face of limited experience and unpredictable future situations, GMO technologies are on the defensive, opening the field for uncertainty, speculation, and stereotypical thinking.

Kahneman and Tversky [41] use prospect theory to argue that humans weight the likelihood of the effects of events according to the size of the perceived outcome rather than objective evidence of the chances of such events occurring. People are also more heavily influenced by possible losses than by even more than equivalent-sized gains. Still, it does not give an answer to the question of why GMO technologies and the use of GM crops for food production are so strongly opposed by societies. There is another theory, which adds a different dimension-resistance to change supported by "confirmatory bias" in the weighting of evidence (only believing evidence that supports existing positions) and the need to conform to peer pressures. Although this theory applies to managing change in a business environment, it may explain the attitudes towards genetic modifications in agricultural production quite well. People may resist change for many reasons—those most relevant to GMOs are probably fear of the unknown, lack of information, misinformation, and, to some extent, lack of perceived or demonstrable benefits. More egalitarian societies have lost trust in experts of all kind, so that constraints based on command structures are rejected and have not been sufficiently replaced with alternative approaches based on two-way discussions, shared and open information, and greater representation in decision making bodies. This is very much the case of GMO, including "lack of benefits", which may be considered negligible or non-existent for the richest societies, mainly in Europe, where the opposition to GMO is the strongest.

As a consequence, national and trans-national policies on GMO technologies that refer to the precautionary principle set in the Cartagena Protocol [42], effectively prevent introducing the cultivation of GM crops, as is the case in the European Union (EU) [43,44], and many single countries, including those that are developing and who suffer from malnutrition.

Ridley [45] provides an example of biotech potatoes modified by the German company BASF in 2005 to produce more of a type of starch that is useful for papermaking and other industrial processes [46]. It illustrates "how impenetrable the EU's regulatory thicket is". Although this GM potato was approved by the European Food Safety Authority (EFSA),

the European Commission blocked this approval referring to the precautionary principle. Irrespective of a second EFSA evaluation confirming the crop was safe, after an intervention by the Hungarian government the BASF application was rejected. Hungary's government intervened on behalf of green pressure groups, pointing out (Kafka-like) that the EU had based its approval on the first EFSA ruling instead of the second one, even though the two rulings were practically identical [45]. BASF consequently stopped the commercialisation of GM products for the European market and moved to the United States.

The history of vitamin A enriched Golden Rice, developed in the 1990s by Potrykus and Beyer Much is more spectacular and symptomatic [47–51]. Vitamin A deficiency, common in Asian and some African countries with a rice-based diet, may cause blindness, and takes the lives of one to two million people annually [50]. Although Golden Rice is a solution to overcome this problem of vitamin A deficiency [52], its adoption was hampered by extraordinarily effective "anti-GMO crop campaigns, and especially anti-Golden Rice campaigns" [17].

Golden Rice differs from white rice, only in that it contains β-carotene, that is, provitamin A, which the human body converts to vitamin A. Golden Rice contains no vitamin A itself. So, the question about safety relates principally to β-carotene, which is ubiquitous in a balanced human diet and the environment [51]. Considering the above, it is difficult to understand the lack of political will and precautionary attitudes that restrain the introduction of a safe and life-saving GMO solution. Only after 20 years after its creation was Golden Rice approved for cultivation and consumption in the Philippines, the first Asian country to do so [53]. Hopefully, this may prove a turning point in the frustrating saga of Golden Rice [17].

The story of Golden Rice is deeply concerning. This is not a story of incompetence and ignorance, but of an antediluvian hostility to science and technology. In the end, though, the evidence in favor of Golden Rice proved absolutely overwhelming [54].

To sum up, the proponents of GMOs see genetic engineering as a tool to mitigate the consequences of climate change, to protect the environment, to increase the productivity of land, and to provide healthier food in many regions of the world.

Opponents fear the threat posed by human interference in natural mechanisms and the eternal laws of nature, which are essential to all life on planet Earth. It can be assumed that the concerns arise mostly from the lack of knowledge in fields such as genetics; the biological and chemical mechanisms of the human body, especially of food digestion and reproduction; and the methodology of genetic modification. In addition, the fact that the first GMO crops were created by agri-business companies should not be neglected, because it laid down the basis for the amalgamation of fears due to the use of the GMO technology and a suspicion of the commercial motivations and market power of the developers. "Activists motivated by these suspicions were successful in getting the 'precautionary principle' incorporated in an international treaty which has been ratified by 166 countries and the European Union—The Cartagena Protocol" [17], which effectively restricts the adoption of GMO crops and strengthens anti-GMO attitudes.

It seems, therefore, that social awareness of the benefits of the introduction of products made by biotechnology is an important element in the process of the adoption of new technologies, including GM foods. This discussion will continue and will certainly have a long and stormy course. Nevertheless, public participation in decision-making regarding issues related to biotechnology is especially important, but difficult to achieve, because of the complex and scientific nature of the issues relating to GMOs. This is one of the reasons for a low level of understanding of basic GM technology principles by consumers, according to the results of studies conducted in China [55], as well as in the USA, Japan, and several European countries (Poland, Italy, Turkey, and Latvia), as reviewed by Wunderlich and Gatto [56].

Education and communication are the methods that, in the process of managing change, are commonly used in situations where there is a lack of information or when information is inaccurate. Although it may be time consuming and costly, once persuaded,

people most often accept the change. In the case of GMO technologies in crop production and GM food, it seems that a relatively small group of scientists and NGOs advocating for the adoption of GMO are the only parties fighting the myths and stereotypes that arose around GMO technologies. Educating and communicating would be a solution; however, policy makers who should take this responsibility are reluctant to act against strong beliefs of their societies, and tend to share the popular views of GM food opponents. Passing regulations restricting the cultivation of GMO crops in fact supports the anti-GMO lobby, as the evidence of ongoing experience is denied.

The aims of this paper are to describe the beliefs found about GMO foods by Polish consumers, producers, and students (renowned for challenges of conventional wisdom) contrasted with the views of GMO scientific specialists. The information created is seen as a contribution to the ongoing debate within Poland and elsewhere, which seeks to openly weigh according to publicly agreed criteria the trade-offs from applying a new and powerful technology in food production. The difficulties of achieving this are centered around perceived risks, strong emotions, logical errors, and biased selection of facts (sometimes for political ends), as enumerated by a compendium of papers published by The Royal Society (1992, Risk Analyses, Perceptions, and Management).

## 2. Materials and Methods

Three surveys were conducted in the years 2014–2018 by students at the Warsaw University of Life Sciences (SGGW) in order to assess the level of knowledge about GMO crops and attitudes towards GM foods in Poland. The original questionnaire constructed by P. Kubisz for the first study ("Public opinion towards genetically modified food", diploma thesis, SGGW, International MBA Program in Agribusiness Management", Warsaw 2014) was used for all the categories of respondents in the following surveys. A convenient sampling approach in the selection of respondents was used. A group of food consumers was contacted through social media. Students of Biotechnology, Economics, Food Technology, and Forestry at the Warsaw University of Life Sciences were approached during their study courses. Farmers from two regions of Poland (Swiętokrzyskie and Opolskie) were surveyed during training sessions on environmental aspects of the Common Agricultural Policy of the European Union. Altogether, 578 respondents were surveyed (Table 1).

**Table 1.** Demographic characteristics of the sample of respondents.

| | Consumers | | Students | | Farmers | | Total | |
|---|---|---|---|---|---|---|---|---|
| | Number (*n* = 219) | % | Number (*n* = 231) | % | Number (*n* = 128) | % | Number (*n* = 578) | % |
| **Gender** | | | | | | | | |
| women | 121 | 55.3 | 162 | 70.1 | 29 | 22.7 | 312 | 54.0 |
| men | 98 | 44.7 | 69 | 29.9 | 99 | 77.3 | 266 | 46.0 |
| **Age (years)** | | | | | | | | |
| Below 25 | 101 | 46.1 | 231 | 100 | 8 | 6.3 | 340 | 58.8 |
| 26–40 | 76 | 34.7 | - | - | 34 | 26.6 | 110 | 19.0 |
| 41–60 | 36 | 16.4 | - | - | 78 | 60.9 | 114 | 19.7 |
| 61 and more | 6 | 2.7 | - | - | 8 | 6.3 | 14 | 2.4 |
| **Education** | | | | | | | | |
| Primary or lower secondary | 5 | 2.3 | 0 | 0.0 | 44 | 34.4 | 49 | 8.5 |
| Upper secondary edu-cation (high school) | 19 | 8.7 | 98 | 42.8 | 58 | 45.3 | 175 | 30.4 |
| Higher education (Bachelor and above) | 195 | 89.0 | 131 | 57.2 | 26 | 20.3 | 352 | 61.1 |
| **Financial standing** | | | | | | | | |
| Not satisfying | 45 | 20.5 | 44 | 19.0 | 38 | 29.7 | 127 | 22.0 |
| Satisfying | 90 | 41.1 | 90 | 39.0 | 65 | 50.8 | 245 | 42.4 |
| Good | 84 | 38.4 | 97 | 42.0 | 25 | 19.5 | 206 | 35.6 |

Source: own elaboration.

Data from the survey presented in this paper have been compared with the answers to the same question by five biotechnology experts from Polish academic institutions.

## 3. Results

Answers to the questions in Table 2 reveal that benefits from using GMOs in agriculture and food production score quite well, although some respondents were confused.

**Table 2.** To what extent does the introduction of GMO serve the following purposes (within the scale 1 = not beneficial at all, to 5 = highly beneficial).

| Purpose | Consumers (*n* = 219) | | Students (*n* = 231) | | Farmers (*n* = 128) | | Sample (*n* = 578) | |
|---|---|---|---|---|---|---|---|---|
| | **Mean** | **Σ** | **Mean** | **σ** | **Mean** | **σ** | **Mean** | **σ** |
| increased resistance to drought | 3.85 | 1.18 | 3.90 | 1.10 | 3.53 | 1.26 | 3.80 | 1.17 |
| higher yields of crops | 3.91 | 1.17 | 4.27 | 0.92 | 3.58 | 1.28 | 3.98 | 1.13 |
| resistance to chemicals | 3.52 | 1.25 | 3.51 | 1.08 | 3.40 | 1.26 | 3.49 | 1.19 |
| cheaper food | 3.46 | 1.29 | 3.45 | 1.23 | 3.10 | 1.47 | 3.37 | 1.32 |
| resistance to pathogens | 3.93 | 1.13 | 3.79 | 1.09 | 3.62 | 1.31 | 3.81 | 1.16 |
| better food quality | 3.50 | 1.30 | 3.70 | 1.17 | 3.05 | 1.41 | 3.48 | 1.30 |
| less hunger in the World | 2.98 | 1.39 | 3.09 | 1.30 | 3.27 | 1.36 | 3.09 | 1.36 |
| cheaper feed for livestock | 3.49 | 1.16 | 3.27 | 1.12 | 3.31 | 1.38 | 3.36 | 1.20 |
| production of medicines | 3.08 | 1.17 | 2.76 | 1.12 | 2.83 | 1.39 | 2.90 | 1.21 |

Σ—standard deviation. Source: own elaboration.

The mean values of the scores for each benefit category at about 3.5 (on a 1 to 5 scale) were reasonably supportive; however, as the standard deviations indicate, opinions on the potential benefits from GMO crops varied widely in the sample (coefficient of variation ranging for the whole sample from about 29 to 47). "Students" were slightly more positive about the potential benefits than "consumers", while, surprisingly, farmers were the most reluctant group. "Higher yields of crops", "resistance to pathogens", and "increased resistance to drought" received the highest scores, while "reducing hunger in the World" and the use of GM technologies to "produce medicines" were assessed on average as the least likely advances.

An indication of the degree of polarisation among respondents was achieved by asking questions about the strength of view for each potential benefit, as captured by the proportion of answers containing statements about negative and positive certainty. The differences are shown in Figure 1.

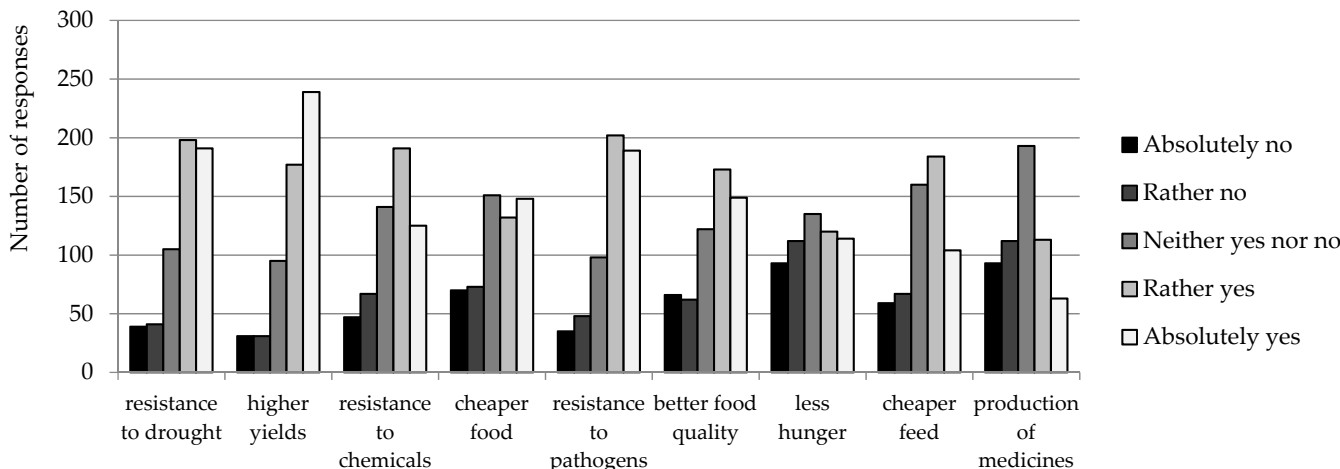

**Figure 1.** The effects of the specific plant genetic modifications in the opinions of respondents. Source: Own research.

The results from the "absolutely not" answer are likely to be strongly held. The number of these respondents as a proportion of the total ranged from less than 10 for most

of the benefits, but doubled for medical advances and reduced hunger. The converse responses of strong support were mostly higher, except where more negative views occurred. This information is potentially most useful for targeting future research and educational campaigns about the reasons for the attitudes of negative "relatively small subgroups". What is not known at this stage is if the extreme answers are for the same respondents across all categories. If this is not the case, i.e., the responses are heterogeneous, it can be surmised that answers were not consistent for each characteristic and would indicate a large amount of confusion. Even so, the findings do suggest that resistance to GMOs is likely to be strongly held by a small number of people.

A measure of the general divergence of opinion was achieved by comparing the mean respondent scores with the mean score in each dimension from the responses of five scientific experts, both expressed in relative terms. The comparisons were informative in revealing much smaller differences of view between lay persons, and a majority scientific one for the husbandry and economic benefits of GMOs, and the much larger dissonance of the anticipated adverse health effects.

The experts were almost unanimous in their assessments. The expert responses are shown as a straight red line at a level of 100 in the figures. Respondents' views measured with this approach were too optimistic for the benefits of extra crop production, but too pessimistic for most of the consequences, revealing evidence of possible inconsistencies in reasoning. Both observations indicate a basic lack of knowledge and understanding in society. The findings will also be useful for the targeting and defining of future research into relevant subgroups of opinions. There did not appear to be great differences in divergence between the three types of respondents, which is surprising for the student group, who typically could be expected to be better informed and more concerned.

Opinions regarding the positive effects of the genetic modifications of plants indicate that respondents are lacking information and also have rather poor knowledge about the real benefits. The share of truly positive answers (absolutely yes, rather yes) ranged from about 31 (production of medicines and vaccines) to 78 (increasing crop yield), although each of the specified futures according to the experts recieved an absolute yes (100 of answers). This indicates that the opinions of respondents were built from stereotypes, not knowledge.

Similar conclusions may be drawn from the analysis of answers to the question about some of the claimed features of GM food, although the percentage of answers that can be considered irrational is slightly less (Table 3 and Figure 2).

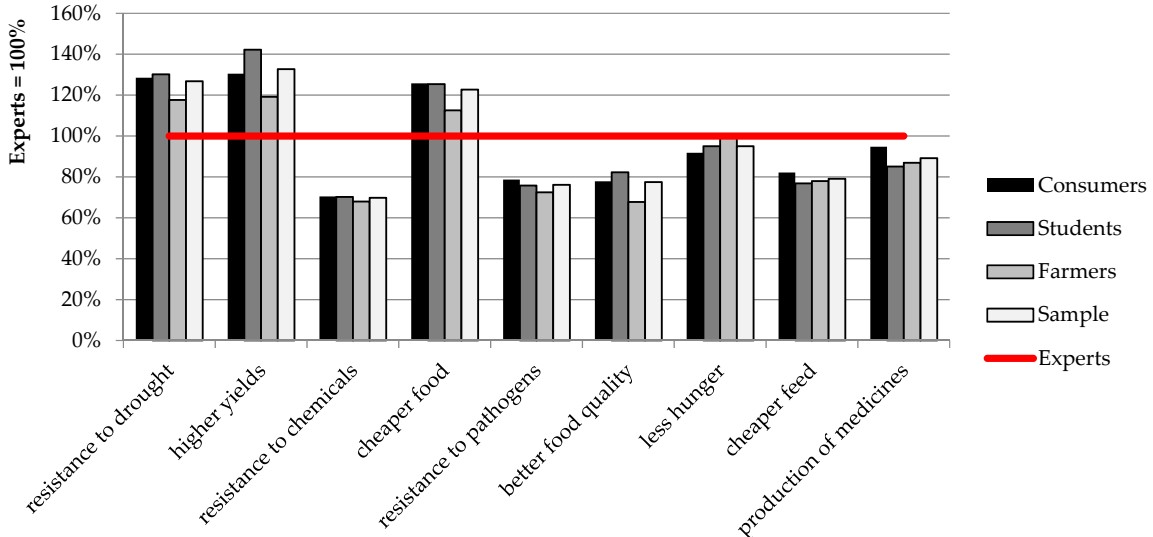

**Figure 2.** Comparison of respondent and expert assessments of the potential effects of introducing GMO technologies in crops. Source: Own research.

**Table 3.** To what extent do you agree with the following statements regarding GM foods?

| Statement GM Food: | Consumers (*n* = 219) | | Students (*n* = 231) | | Farmers (*n* = 128) | | Sample (*n* = 578) | |
|---|---|---|---|---|---|---|---|---|
| | Mean | Σ | Mean | σ | Mean | σ | Mean | σ |
| Is difficult to digest for humans | 2.63 | 1.12 | 2.54 | 1.20 | 3.17 | 1.38 | 2.71 | 1.24 |
| Is allergic for humans | 3.11 | 1.22 | 3.33 | 1.26 | 3.27 | 1.31 | 3.23 | 1.26 |
| Causes the degradation of the internal organs | 2.80 | 1.20 | 2.76 | 1.27 | 3.18 | 1.26 | 2.87 | 1.25 |
| Causes infertility | 2.74 | 1.19 | 2.61 | 1.29 | 3.14 | 1.31 | 2.77 | 1.27 |
| Changes hereditary features of human organisms | 2.89 | 1.19 | 2.60 | 1.24 | 3.02 | 1.30 | 2.80 | 1.25 |
| Was introduced to generate profits for "GMO corporations" | 3.83 | 1.30 | 3.84 | 1.12 | 3.42 | 1.48 | 3.74 | 1.28 |
| Is more durable and fresh | 3.67 | 0.96 | 3.78 | 1.02 | 3.19 | 1.39 | 3.61 | 1.11 |
| Is cheaper than food without gmos | 3.38 | 1.12 | 3.32 | 1.36 | 3.25 | 1.48 | 3.33 | 1.30 |

σ—standard deviation.

Some of the answers demonstrate the power of myths in forming beliefs and the exaggeration and speed of their dissemination through the Internet and social media, such as GMO food is "allergic for humans"(44 answered absolutely and rather yes) or "causes the degradation of the internal organs" (28 answered absolutely and rather yes). A larger share of answers of "neither yes or no" indicates that respondents have insufficient knowledge and/or are less emotional about GM technologies. The answers are not clear on whether GMO is either "good" or "bad". A comparison with the degree of support for the general benefits of GMO (Table 2 and Figure 1) shows that the polarization of views about the consequent properties of GMOs (Table 3 and Figure 3) is much greater—a possible case of more extreme views being more widely supported and with more confidence. A classic case of "confirmatory bias".

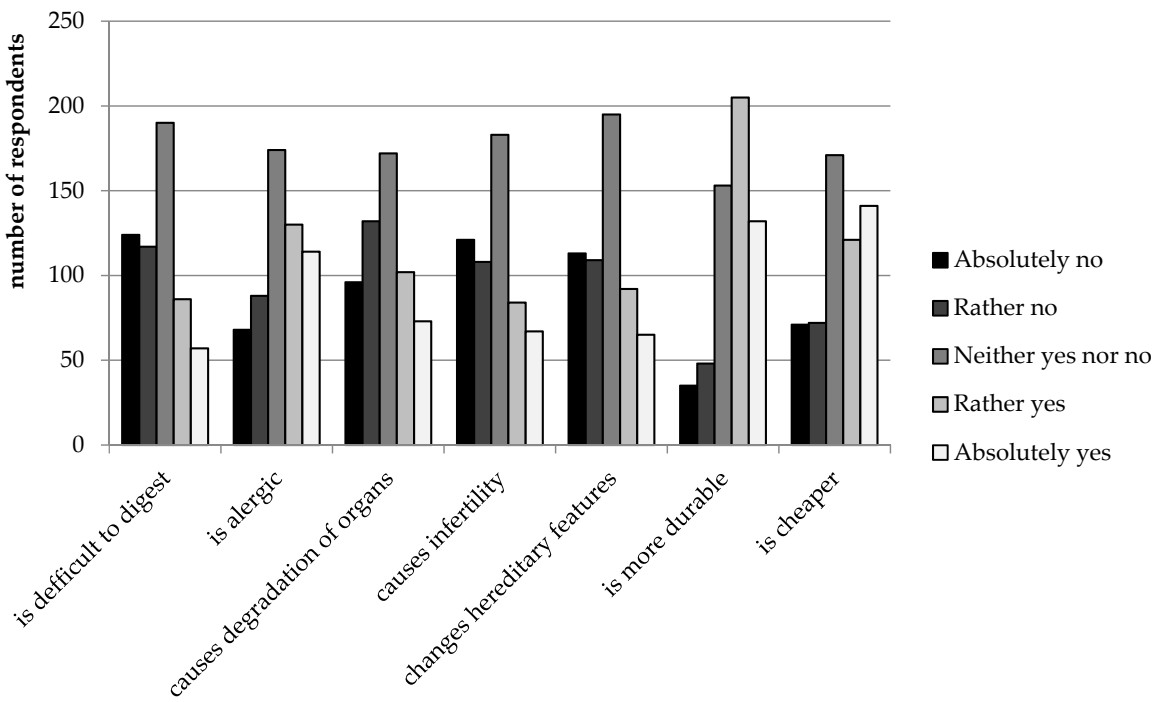

**Figure 3.** Share of positive and negative opinions on specific features and perceived consequences of GM food. Source: Own research.

The relatively small differences between the categories of respondents in their assessments indicate that a rather low level of awareness and stereotypes about GMOs are prevalent in all branches of Polish society (Figure 4). This conclusion is also supported by the evidence in Figure 4 of the mean weighting given to GMO health characteristics by respondent types as a large multiple of the experts base scores.

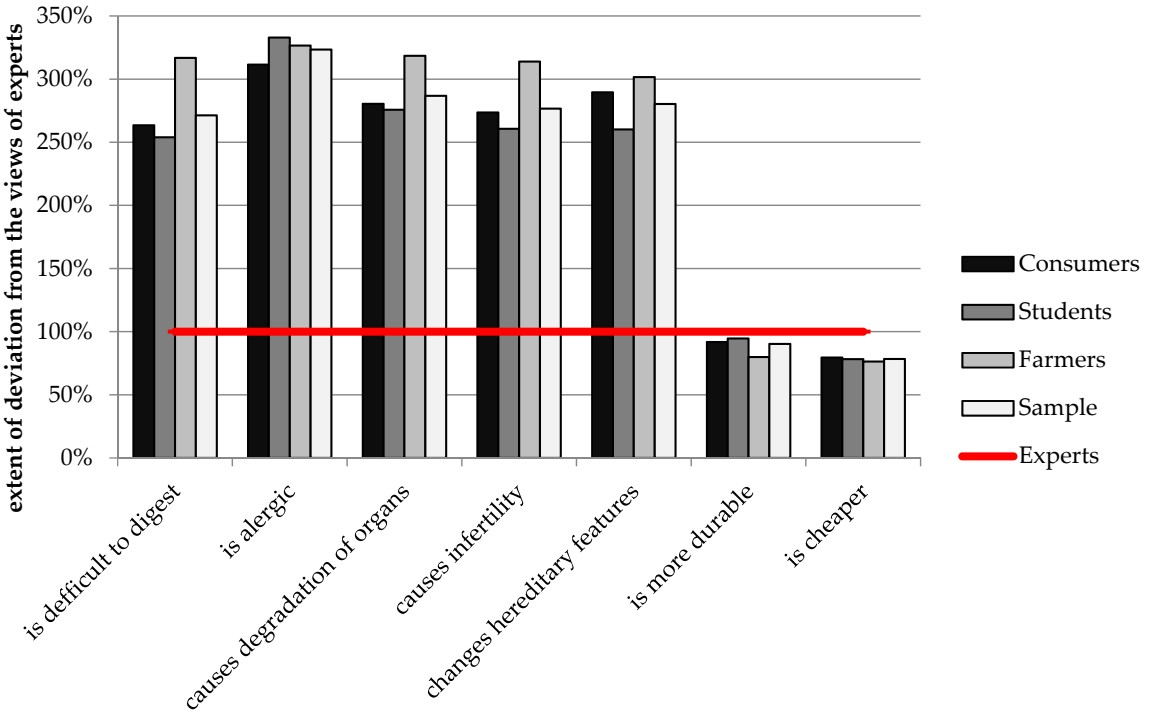

**Figure 4.** Comparison of the opinions on the features and consequences of GM food between different categories of respondents. Source: Own research.

The negative attitude of the public towards GMOs also shapes the level of acceptance for non-food uses (Table 4).

**Table 4.** The level of acceptance for non-food uses of GMO.

|  | Consumers (*n* = 219) | | Students (*n* = 231) | | Farmers (*n* = 128) | | Sample (*n* = 578) | |
|---|---|---|---|---|---|---|---|---|
|  | **Mean** | **σ** | **Mean** | **σ** | **Mean** | **σ** | **Mean** | **σ** |
| Medicine/Pharmacology | 3.48 | 1.38 | 3.56 | 1.43 | 2.48 | 1.53 | 3.29 | 1.50 |
| Feed industry | 2.93 | 1.34 | 2.82 | 1.41 | 2.67 | 1.53 | 2.83 | 1.41 |
| Environmental protection | 3.37 | 1.37 | 3.18 | 1.40 | 2.50 | 1.54 | 3.10 | 1.46 |
| Biofuel's production | 3.96 | 1.26 | 4.18 | 1.09 | 3.51 | 1.52 | 3.94 | 1.29 |
| Paper industry | 3.84 | 1.29 | 3.92 | 1.22 | 2.48 | 1.58 | 3.57 | 1.36 |

Source: Own research.

There is no rational reason for not using genetic modifications when growing crops that may be highly effective and safe when used in all industries. It seems that objections are thus not against the technology as such, but are more connected with its consequences, which, in the case of food, are perceived to have adverse health effects.

## 4. Discussion

There are several studies showing rather low levels of public understanding regarding what GMOs really are, as well as varied attitudes towards GM food.

McFadden and Lusk [57], who surveyed a representative sample of USA citizens, found that about 40% of respondents rated themselves as "very or somewhat knowledge-

able" and only 34% believed the GM food was safe, which is noticeably contrasting with 88% of the scientist members of the American Association for the Advancement of Science who believe GM food is safe to eat [58]. Another survey of USA consumers revealed that 54% of the respondents knew very little or nothing about GMOs, what is more, 25% had never heard of GMOs [59].

Lack of knowledge about GMOs often affects the attitudes of consumers towards GM food [60,61]. In the survey conducted in 2010 in all EU countries, "genetically modified organisms found in food and drink" was a "worry" for 66% of respondents (ranging from 46% in Ireland, Malta, Sweden, and the United Kingdom, to 8% in Greece and Lithuania), while consumers were less worried about "food poisoning from bacteria like salmonella in eggs or listeria in cheese" (62% in EU27), ranging from 23% in Sweden to 85% in Cyprus [62]. A more recent survey shows, however, that GMOs dropped from number four to number eight on the 2010 list of food safety concerns in EU society, which, to some extent, is a likely a result of new worries, such as micro plastics [63].

According to a report "Poles and GMOs" [64], about 65% of respondents did not understand the true meaning of the GMO acronym, and most (60.7%) supported the statement that the adoption of genetically modified crops is beneficial mainly for companies introducing GMOs. Most respondents (57.1%) expressed the view that placing GMOs on the market primarily serves GMO companies and that it has a negative impact on human health [64]. The respondents were inclined, however, to agree that, due to genetic modifications, organisms acquire valuable features such as better taste and smell. A low level of understanding of GMOs by Polish respondents was also reported by Krzysztofik [48] and Kosicka-Gebska and Gebski [60,65]. Similarly, the results of a survey of students of agriculture showed that 50% were convinced that GM foods create threats to human health [66]. In another study conducted among students of the Biotechnology Department of Poznań University of Medical Sciences, students expressed greater support for genetic modification and were more willing to buy and consume such products. Opinions among the dietetics students were more divided. The level of knowledge about GMO in both groups of students was assessed as being high [61].

The results from this study confirm many of the previous findings in the literature. They show broad support for the general benefits of GMOs, which did not vary between the three groups of respondents surveyed (consumers, farmers, and students; Table 2), but noticeably differed with the experts' views (Figure 2). The spread of opinions depicted in Figure 3 shows a small proportion are confirmed pessimists, but they do not have a majority over those who are surer, and the greatest proportion who support the technology although with some doubts. The doubters are more frequent for possibly more surprising advances such as resistance to chemicals, and for those that require other conditions to bring them about, such as cheaper food.

The possible effects on human health are the greatest concern (Table 3 and Figure 3), as in the literature, but even so, the beliefs are not held strongly by most respondents. Allergies are the greatest concern. The health effect beliefs are widely different from those of experts (Figure 4).

The legislation of applying the precautionary principle within the EU to ban the technology has apparently been driven mainly by the imagined fears of a minority of people. A consequence of the ban is that there is no experience to draw on, other than that from other countries. This state of affairs could explain the similarity of the responses between farmers, consumers, and students—all of which rely on secondary information. The focus just on health fears while ignoring the better supported benefits of GMOs in ameliorating climate change, pollution, biodiversity, and food insecurity justifies an urgent evaluation of the trade-offs ignored by the current ban.

## 5. Conclusions

It is clear that, in Poland, as in many other countries, there are two groups of consumers—the first who accept GMOs and the second, smaller group, who strongly oppose them as a perceived threat, mainly to human health and the environment.

There are many myths about biotechnology, widely spread by organizations acting against genetically engineered products for ideological, political, or economic reasons. These myths, which tend to be extreme and firmly held, may be held by only a small part of the population. Unfortunately, the lessons from seminal reports [24], such as those of the adoption by a large number of Chinese farmers some 30 years ago regarding insect resistant cotton varieties that demonstrated enhanced farm profitability, reduced the rates of insecticide poisoning and lowered prices, and were therefore better for consumers, are overlooked. More recent studies [67] stress the importance of competition between state and private sector companies, which gives farmers greater choice and promotes more investment in future research and development, as well as improved seed supply mechanisms. Legislation (instead of more market led competition) that tries to curb trade and the activities of "multinationals" often produces sub optimal behavior. On the other hand, there are now a number of scientific reports and facts explaining the issues of GMOs, putting an emphasis on the usefulness of genetic engineering in modern agriculture and food production. The discoveries of today's science fascinate not only scientists, but also society. However, information about GMOs is not presented in a way that is fair and understandable for the average consumer. Most people's preferences are driven by emotion rather than facts. Attractive reports are more interesting and attention grabbing than hermetic scientific opinions.

The exploitation of the great opportunities posed by biotechnology does require a higher level of public knowledge and education. Consumers will decide about the future of biotechnology by their willingness or not to purchase its products.

It is clear that the adoption of technology depends as much on gaining consumer and other stakeholders' confidence as it does on advances and applications in basic science. How might this be done? This study provides some starting points, such as the revelation of beliefs about the collateral impacts on human health, risks to ecosystems, and the motives of multinational companies as being the more important impediments to changing acceptance than perceptions of potential benefits. The variations in these beliefs are more or less the same across the three categories of respondents surveyed (consumers, farmers, and students), especially when compared with views of experts.

In the meantime, market responses should focus on enhanced labelling and differential pricing of food produced in traditional ways, including organic and with or without GMOs. An example is due diligence contracts, which include price premia issued by supermarkets for guaranteed provenance, which is validated by inspections of suppliers' prescribed activities.

Market led approaches point to promising innovations in research methodology akin to the testing and administration of COVID measures, but also from marketing and consumer research studies and economic development tools. "Big data" digital methodology for representative samples of more population types and comparisons with existing data bases such as those derived from the European Social survey(s) in order to reflect societal heterogeneity makes it possible to manage and cross correlate exceptionally large data sets. Smaller studies such as those reported here are still valuable for scoping purposes and producing "first cut" understandings of what is going on. A specific complementary tool, now 35 years old, is the late Edward de Bono's book *Six Thinking Hats*, which could be used to handle the emotions involved in focus groups [68]. Major advances have also occurred in behavioral theory [69] and its application in farmer actions [70].

Randomized control trials are also used extensively to improve the quality of interventions in economic actions so as to allay poverty in many developing countries, whereby different groups are compared in their actual responses to interventions compared with other groups who are not beneficiaries (see [71]). There may be resource and organizational

challenges in using such studies, but the case for them is that they can demonstrate actual responses that often challenge contemporary wisdom—an approach that may be necessary given the firmness of so many of the erroneous views about GMOs. Fortunately, there are also opportunities to demonstrate the considerable experiences of countries already using GMOs.

In the context of important topics such as global warming, biodiversity decline, pollution, and inequality in situations of growing poverty, it is most unsatisfactory to come to conclusions about GMO by considering them as a single issue. If they are not to be used, the question of applying the precautionary principle needs careful consideration, as the risks of denying their use in context could far exceed those of using them.

Scientific advances cannot be undone, and consumer's fears determine their choices. Returning on a wide scale to old technologies that require extensive farming methods and traditional food processing appears to be impossible because of limited land resources that are insufficient to meet the rising demand for food for a larger and wealthier world population.

Nowadays, we are experiencing an increasingly accelerating scientific revolution that is difficult for the average consumer to follow. The scope and pace of changes in various areas of life—such as the introduction of 5G wireless technology or unprecedented achievements in the development of the latest generation of anti-COVID vaccines—is difficult to digest. By imposing the natural fears of the unknown of every human being, the caution of policy makers in making controversial decisions, possible conflicts of interest, and the often-incomprehensible activity of opponents of innovative solutions, a climate is created in which it is easy to succumb to emotions and ignore difficult to grasp scientific messages.

Humanity needs GMOs—not only for food production, but for many other uses in various non-food industries. However, the statement "there will be no crop biotech revolution unless scientists and consumers learn to talk to each other" [72] is undoubtedly very true. This implies a need for emotionless discussion based on well documented scientific grounds and through lessons of demonstrable experience on issues of concern such as those featured in this paper. The public should not get the impression that GMO technologies are being introduced through the back door. This requires strengthening communication with the public through accessible explanations of GM technologies and the benefits they provide.

While shedding a light on the noticeable divisions within society from basically two groups of consumers—those who are strongly opposing and those who accept genetic crop modification—we are aware of the limitations, which can guide future research. First of all, our sample is not well balanced. In particular, the groups of consumers and students are very homogenous, which does not allow for a more in-depth analysis of attitudes considering different demographic characteristics. It should be noted that all of the respondents were consumers, and by differentiating them into three groups, it did not add to our understanding, at least in average terms.

**Author Contributions:** Conceptualization, P.K. and E.M.; methodology, P.K., E.M. and K.P.; software, P.K. and K.P.; validation, G.D. and E.M.; formal analysis, K.P. and P.K.; investigation, K.P. and P.K.; data curation, K.P.; writing—original draft preparation, P.K. and E.M.; writing—review and editing, G.D.; visualization, K.P. All authors have read and agreed to the published version of the manuscript.

**Funding:** This research received no funding.

**Institutional Review Board Statement:** Not applicable.

**Informed Consent Statement:** A consent was waived—all respondents voluntarily replied to questions, there were no sensitive private data collected, identification of persons responding is not possible.

**Data Availability Statement:** Data collected in the surveys are deposited in an Excel file at the Warsaw University of Life Sciences.

**Conflicts of Interest:** The authors declare no conflict of interest.

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
