# Peer review of "Facts and Myths about GM Food—The Case of Poland"

_agriculture, doi:10.3390/agriculture11080791_

Round 1

Reviewer 1 Report

This study examines the consumer perceptions of  GM products and compares them against scientific, objective views.

 3 groups of respondents: consumers, students, farmers take part in the study.

The introduction shows the problem under investigation but it is not clear what is the aim of this study. The authors should include the research question(s) or aim and intended study contributions.

The literature review takes an objective view of GMOs. The consumer fear and hesitation against GM products is described well. The author(s) could develop this section further; there is  discussion of GM in general including policy perspectives which, albeit interesting, not directly related to this study. Another point is that the section 4. Consumer’s knowledge and attitudes is mostly about consumer knowledge; there is little discussion of the topics investigated via the survey, e.g., quality, productivity, hunger. The authors may expand this section further and structure it around the survey topics.

Methods lack a description of how the survey was contacted; questionnaire is also missing; more details are needed about the participant demographics. Further, how the authors got the interview questions? From previous studies and which one?

Findings are interesting. I am not sure without seeing the questionnaire, if there are data available to do cross-tabulations e.g. by age and education. In figure 2, experts are 100% in all dimensions. Why? It needs further explanation.

In conclusion, it says:

It is clear, that in Poland, as in many other countries there are two groups of consumers: the first who accept GMO’s and the second who strongly oppose them as a perceived threat to human health and the environment.

This is an interesting finding. It would be great if possible to run a cluster analysis to uncover these 2 clusters and their characteristics. If not, this should be noted as a limitation. Like any technological innovation, GM products may be characterised by the innovation diffusion groups (5-stage innovation adoption): innovators, early adopters, late adopters, laggards. There is no mention to it although it seems that there is such separation among consumers

Reviewer 2 Report

  1. The abstract didn't show enough information, especially the part of the results. The ‘Objective- method - result - conclusion’ pattern is suggested to be followed.
  2. In line 37, ‘According to numerous studies……’,Why did the authors refer to only one document? From line 42 to line 50, publications that can support the authors’ opinion are necessary.
  3. In part 2, which is for the genetically modified food and which is against? The structure of this section is not clear enough. Similarly, the same goes for parts 4 and 5.
  4. It is not very close of the logical connection between parts 4 and 5. Why and how the three categories of respondents are selected?
  5. From line 302 to line 305, the past studies quoted in the literature review are still required.
  6. The survey wasn't complicated, and it is quite confusing that four years were spent. It is recommended to give sample descriptive statistics before verifying the survey results. Do a reliability and validity test if the authors can.

Round 2

Reviewer 1 Report

the authors revised the manuscript based on my recommendations. I read their responses and the revised manuscript.  Most comments were addressed adequately; some recommendations were not followed but the authors justified their opinion; although I dont fully agree with some decisions, I cannot suggest further revisions since it is the authors' study and research design to go on with it.

Reviewer 2 Report

Thank the authors for the efforts to improve the quality of the paper.